# Defect-Mediated Energy Transfer Mechanism by Modulating Lattice Occupancy of Alkali Ions for the Optimization of Upconversion Luminescence

**DOI:** 10.3390/nano14231969

**Published:** 2024-12-07

**Authors:** Rongyao Gao, Yuqian Li, Yuhang Zhang, Limin Fu, Luoyuan Li

**Affiliations:** 1Laboratory of Advanced Light Conversion Materials and Biophotonics, School of Chemistry and Life Resource, Renmin University of China, Beijing 100872, China; rygao@ruc.edu.cn (R.G.); liyuqian@ruc.edu.cn (Y.L.); zhangyuhang@ruc.edu.cn (Y.Z.); 2The Eighth Affiliated Hospital, Sun Yat-sen University, Shenzhen 518033, China

**Keywords:** alkali ions, upconversion, energy transfer, imaging in cells

## Abstract

The performance optimization of photoluminescent (PL) materials is a hot topic in the field of applied materials research. There are many different crystal defects in photoluminescent materials, which can have a significant impact on their optical properties. The luminescent properties and chemical stability of materials can be effectively improved by adjusting lattice defects in crystals. We systematically studied the effect of doping ions on the energy transfer upconversion mechanism in different defect crystals by changing the matrix alkali metal ions. Meanwhile, the influence mechanism of crystal defect distribution on luminescence performance is explored by adjusting the ratio of Na–Li. The PL spectra indicate that changing the alkaline ions significantly affects the luminescence performance and efficiency of UCNPs. The change in ion radius leads to substitution or gap changes in the main lattice, which may alter the symmetry and strength of the crystal field around doped RE ions, thereby altering the UCL performance. Additionally, we demonstrated the imaging capabilities of the synthesized upconversion nanoparticles (UCNPs) in cellular environments using fluorescence microscopy. The results revealed that Na_0.9_Li_0.1_LuF_4_–Yb, Er nanoparticles exhibited significantly enhanced fluorescence intensity in cell imaging compared to other compositions. We further investigated the mechanism by which structural defects formed by doping ions in UCNPs with different alkali metals affect energy transfer upconversion (ETU). This work emphasizes the importance of defect regulation in the ETU mechanism to improve the limitations of crystal structure on the luminescence performance and promote the future application of upconversion nanomaterials, which will provide important theoretical references for the exploration of high-performance luminescent materials in the future.

## 1. Introduction

Luminescent materials are an important component of the preparation of photonic devices, optical sensors, and detectors, widely used in high-tech fields such as display lighting, biotechnology, optical anti-counterfeiting, and optical sensing [1,2,3,4,5,6,7,8]. The performance optimization of luminescent materials has always been a hot topic in the field of applied materials research. In the process of crystal synthesis, the irregular arrangement of crystal cell atoms may cause the generation of crystal defects. Photoluminescent materials are mainly applied in the form of polycrystalline materials, in which crystal defects may occur at grain boundaries, phase boundaries, and surfaces, and are prone to dislocations, vacancies, etc. [9,10]. These defects will have a certain impact on the optical properties of optical crystals, such as photorefractive properties, light damage resistance, and refractive index [11,12,13]. Especially, point defects have a significant impact on their luminescent properties, making them the research focus of photoluminescent materials. It can effectively improve the luminescence performance and chemical stability of materials by introducing lattice defects into crystals. The distribution of defects could be regulated by adjusting the element ratio, co-doping sensitizers, and substituting with ions of the same main group or near radius. It is a remarkably attractive research field because effectively regulates defects for designing materials with the required performance, which will provide important theoretical references for the subsequent exploration of high-performance luminescent materials [14,15,16,17,18,19,20,21,22].

Rare earth (RE)-doped upconversion nanoparticles have attracted widespread attention due to their superior physicochemical features such as large anti-Stokes shifts, sharp-band emissions, tunable emission, long lifetimes, as well as low auto-fluorescence background from biological samples. Traditional molecular probes are always excited in the ultraviolet or visible spectral region, so there are limitations in biological imaging due to auto-fluorescence, photo-bleaching, and photo-damage. RE-doped nanoparticles show unique upconversion luminescence (UCL) properties under near-infrared (NIR) laser irradiation, such as a large anti-Stokes shift, long lifetime, and sharp emission line [23,24,25,26]. Attributed to the complicated energy levels and multiple transitions of RE ions, it is considered to be important for applications such as biosensors, optoelectronic probes, and display devices to manipulate pure color output [1,2,3,7,27,28]. However, the luminescence intensities and quantum efficiency of RE-doped nanomaterials are limited to some extent in promoting future applications. Recently, lots of approaches have been employed to enhance the UCL properties through internal or external adjustments, such as changing crystal phase, doping-assisted sensitizer, and deactivating surface defects by core/shell structure [16,17,18,19,20,21,29,30]. However, it is still a formidable challenge to further increase the luminescence properties of RE-doped nanomaterials.

It is well known that the UCL intensities of RE-doped nanomaterials depend on their 4f transition probabilities, which are affected significantly by the crystal field population of sensitizer and luminescence ions. It should be noted that the narrow excitation band covers at most 100 nm for the 4f−4f inner-shell transitions considering crystal field splitting [1]. Numerous efforts have been attempted to overcome the limitation of crystalline structures. Despite these ways to some extent improving the UCL properties, the mechanism of energy transfer in intra-crystal structures is affected by different alkali ions in RE-doped nanomaterials [31,32,33,34]. Therefore, there is a lack of systematic research on the impact of alkali ions on the energy transfer upconversion mechanism in rare earth nanomaterials.

In this work, the influence of doping ions on the energy transfer upconversion mechanism in different defect crystals was systematically studied by changing the matrix alkali metal ions. Meanwhile, the ratio of Na–Li was regulated to explore the mechanism of crystal defect distribution on luminescence performance. We also measured the steady-state photoluminescence (PL) and nanosecond time-resolved PL spectra of these different UCNPs at room temperature. The PL spectrum shows that changing the alkali ions (Li^+^ to Cs^+^) can have a significant impact on the luminescence performance and efficiency of UCNPs. The change in ion radius causes substitution or interstitial changes in the host lattice, which could alter the symmetry of the crystal field around RE-doped ions, thereby altering the UCL performance. We further explored the influence mechanism of structural defects formed by doped ions in the UCNPs of different alkali metals on energy transfer upconversion. This work emphasizes the importance of defect regulation on the ETU mechanism to improve the limitations of crystal structure and promote the future application of upconversion nanomaterials.

## 2. Materials and Methods

### 2.1. Materials

Rare-earth oxides, Lu_2_O_3_ (99.99%), Yb_2_O_3_ (99.99%), Tm_2_O_3_(99.99%), and Er_2_O_3_ (99.99%), were purchased from Foshan Lansu Chemical Industry Company (Foshan, China). NH_4_F (98%) and NaOH (98.5%) were purchased from J&K Chemical (J&K Chemical, Hong Kong, China). LiOH (98%), KOH (90%), and CsOH·H_2_O (99.95%) were purchased from Sigma-Aldrich, St. Louis, MO, USA. Oleic acid (>90%), HCl (37%), and 1-octadecene (>90%) were purchased from Alfa Aesar (Waltham, MA, USA). Cyclohexane and ethanol were supplied by Beijing Chemical Plants (Beijing, China). Hoechst 33,258 (98%) were purchased from Sigma-Aldrich. LysoTracker was purchased from Aladdin (Dubai, United Arab Emirates). RE chlorides (LuCl_3_, YbCl_3_, TmCl_3_, and ErCl_3_) were obtained by dissolving the metal oxides in 10% hydrochloric solutions and then evaporating the water completely. In a traditional procedure, the doping concentration of Yb^3+^ and Er^3+^/Tm^3+^ was fixed at 20% and 2%, and the upconversion nanoparticles (UCNP) with different alkali ions (M: Li, Na, K, Cs) were synthesized by a solvothermal method. Firstly, LuCl_3_ (0.78 mmol), YbCl_3_ (0.20 mmol), and ErCl_3_/TmCl_3_ (0.02 mmol/0.01 mmol) were added to four 250 mL three-necked flasks, respectively; then, 1-octadecene (10 mL) and oleic acid (6 mL) were added followed by heating to 140 °C under the atmosphere of argon to form a transparent solution. After cooling down to room temperature, 2.5 mmol of different alkali hydroxides (LiOH, NaOH, KOH, and CsOH) and 4 mmol NH_4_F were added to these flasks and stirred for 30 min under vacuum, respectively. Subsequently, the solutions were heated to 295 ± 3 °C for 1 h under an Ar atmosphere. At 25 °C, the mixed solution was treated with 2 mL of ethanol and 2 mL of cyclohexane, followed by centrifugation at 12,000 rpm for three cycles to obtain the final precipitated product. Finally, the obtained four kinds of upconversion nanoparticles with different matrices that doping with different alkali ions were dried under 60 °C for 6 h. A549 cells (non-small-cell lung cancer cells, NSCLCs) were grown in RPMI 1640 medium supplemented with 10% fetal bovine serum (FBS, Gibco, Waltham, MA, USA). Cultures were maintained at 37 °C under a humidified atmosphere containing 5% CO_2_.

### 2.2. Experimental Methods

The size and morphology of the nanoparticles were determined at 120 kV using a Hitachi H-7650B transmission electron microscope (TEM) (Hitachi, Tokyo, Japan) and a low- to high-resolution transmission electron microscope. The distribution histograms of particle sizes were derived from the TEM images of more than 200 nanoparticles. The EDX spectra of the samples were acquired using a field emission scanning electron microscope coupled with an energy-dispersive X-ray spectrometer operating at 10.0 kV. The crystal form of the nanomaterial was analyzed by Shimadzu XRD-7000 X-ray diffraction (XRD, Shimadzu Corporation, Kyoto, Japan). The radiation source was the Cu Kα line (λ = 0.1540), the scanning speed was 2°/min, and the scanning range was 10°~70°. The reference data come from the Joint Committee on Powder Diffraction Standards (JCPDS). All the nanomaterials were re-dispersed in ethanol for further spectral testing. The steady-state upconversion luminescence spectra were measured using an FLS980 fluorescence spectrometer (Edinburgh Instruments, Edinburgh, UK), which was excited by an external continuous-wave 980 nm diode laser (Changchun New Industries Optoelectronics Tech Co., Changchun, China, MDL-H-980-5W) scanning from the range of 500 nm to 700 nm. In this experiment, the sample is a dispersion of 1 mg/mL UCNP in cyclohexane, with an excitation power density of 500 mW/cm^2^, and the experimental conditions for each sample are the same. The time-resolved luminescence spectra were collected by fluorescence lifetime spectrofluorometer in multi-channel scanning mode combined with a signal synchronous modulator (974 nm pulse) that was generated by a YAG laser (355 nm, 7 ns, 10 MHz); pumped optical parametric oscillator sets were used as the pump source. For fluorescence microscopy imaging, A549 cells were incubated with the medium containing upconversion nanoparticles (200 μg⋅mL^−1^) for 24 h. The fluorescence signal was collected at 540 ± 30 nm exposed by a 980 nm (200 mW/cm^2^) continuous wave laser and measured by obtained digitally on a Nikon CCD camera (Nikon, Tokyo, Japan). Cell nucleus and lysosome were stained with Hoechst-33258 (Sigma Aldrich, St. Louis, MO, USA) and LysoTracker (Thermo Fisher Scientific, Waltham, MA, USA), respectively. All the tests were recorded at room temperature.

## 3. Results and Discussion

In this work, the UCNPs with different alkali ions (Li, Na, K, Cs) were obtained through the conventional solvothermal method. Firstly, the doping concentration of Yb^3+^ and Er^3+^ was fixed at 20% and 2%, and the UCNPs with different alkali ions were identified as MLuF_4_ (M: Li, Na) and MLu_2_F_7_ (M: K, Cs). Their morphology and particle size were measured through a transmission electron microscope (TEM). The samples showed different shapes (Figure 1a), diamond-shaped for LiLuF_4_ nanoparticles, sphere-shaped for NaLuF_4_ nanoparticles, and hexagon-shaped for KLu_2_F_7_ and CsLu_2_F_7_ nanoparticles. The particle size distributions (Figure 1c) showed LiLuF_4_ of ~25 nm, NaLuF_4_ of ~16 nm, KLu_2_F_7_ of ~19 nm, and CsLu_2_F_7_ of ~23 nm. Energy-dispersive X-ray spectra (EDX) displayed the elemental components of the corresponding nanocrystals (Figure 1c), and the data showed there were differences in the alkali elements of these samples. The result indicated that different alkali ions induced various crystalline forms. To further indicate the crystal phase data, the samples were characterized by X-ray diffraction. Figure 2 shows that the patterns of these samples correspond to the standard cards (JCPDS card No. 27-1251, 27-0726, 27-0459, 43-0504) of the pure hexagonal (β-) phase, respectively. The β-phase nanoparticles had been proven to perform stronger UCL properties than α-phase UCNPs.

To demonstrate diversity caused by different alkali ions, the upconversion luminescence (UCL) properties of these samples were measured by steady-state photoluminescence (PL) spectroscopy under a 980 nm continuous-wave laser irradiation. The MLuF_4_ nanomaterials co-doped with Er^3+^ and upon excitation of Yb^3+^ in the near-infrared absorption band at 980 nm. These emissions can be attributed to ^2^H_11/2_ → ^4^I_15/2_ (520 nm), ^4^S_3/2_→ ^4^I_15/2_ (539 nm), and ^4^F_9/2_ → ^4^I_15/2_ (654 nm) transitions of Er^3+^, respectively [35,36,37,38]. The PL spectra of these nanoparticles doped with 20% Yb^3+^, 2% Er^3+^ and 20% Yb^3+^, 20% Er^3+^ with different alkali ions (Li, Na, K, and Cs) showed the UCL intensities of 540 nm emission significantly stronger than the UCL intensities of 654 nm emission (Figure 3), so that the UCL color mainly displayed green. And these spectra of the samples showed there were significant differences in the ratio of red emission (654 nm) to green emission (540 nm), whereas the UCL intensities decreased gradually with alkali ions of host materials changed from Li^+^ to Cs^+^. The splitting degree of characteristic emissions also changed significantly, especially at 540 nm emissions. These results indicated that the remarkable changes in UCL intensity and the splitting degree of characteristic emissions could be attributed to the host materials with different alkali ions (Li, Na, K, and Cs). And the two kinds of Er^3+^-doped nanoparticles were used to demonstrate that the defect-mediated energy transfer mechanism could be adjusted by the lattice occupancy of different alkali ions, especially for the impact of red light (^4^F_9/2_ → ^4^I_15/2_, 654 nm). As well, the steady-state PL spectra of UCNPs by doping with Yb^3+^ (20%) and Tm^3+^ (1%) were measured according to the same method (Appendix A, ESI). To study these changes induced by different alkali ions, we carried out a more in-depth analysis. Firstly, the important parameters of the ion-radii ratio are 0.414 (2^0.5^–1) and 0.732 (3^0.5^–1). If the radius ratio of positive and negative ions (r^+^/r^−^) is not less than 0.732 (r^+^/r^−^ ≥ 0.732), the coordination number of metal ions is 8 coordination; when 0.732 > r^+^/r^−^ ≥ 0.414, then the coordination number is 6. In these samples, their ion radius was approximately 0.076 nm (Li^+^), 0.102 nm (Na^+^), 0.133 nm (K^+^), and 0.169 nm (Cs^+^), respectively. According to these data,

It could be considered that the coordination forms of Li^+^ and Na^+^ in the UCNP matrix (MLuF_4_) are obviously different from that of K^+^ and Cs^+^ in the UCNP matrix (MLu_2_F_7_), thereby affecting the distribution of the electronic cloud. In the hexagonal UCNPs, the lattice structure consists of one type of anion site occupied by F ions, and two types of cation sites selectively occupied by alkali ions and RE^3+^ ions, resulting in partial electron cloud distortion to accompany the lattice changes. With the ionic radii increasing (Li^+^ → Cs^+^), the electron cloud exhibited dramatic change due to the dipole polarizability increased. It can be reasoned that doping of alkali ions with an obvious difference in ion radii with Lu^3+^ (0.076 nm) in the host lattices should influence the structure formation of upconversion nanocrystals as show in Figure 1, which can further alter the UCL properties. The kinetic curves and fitting lines of the ^1^G_4_ (477 nm) and ^3^H_4_ (800 nm) levels of Tm^3+^ were also explored (Appendix A, ESI).

To further investigate the ETU population and UCL process in these samples with different alkali ions (Li^+^, Na+, K^+^, and Cs^+^), their lifetime decay was measured. The kinetic curves of the ^4^S_3/2_ (540 nm) and ^4^F_9/2_ (654 nm) levels of Er^3+^ have been explored (Figure 4). And the fitting results for the rising component are shown in Appendix A. These kinetic curves were fitted with the bi-exponential equation as the following Equation (1):
*I*_t_ = A_1_*exp* (−*t*/τ_1_) + A_2_*exp* (−*t*/τ_2_)(1)
where τ_1_ and τ_2_ represent the rising and the decay time constant describing the time scales of population and depopulation for corresponding energy levels, respectively. A_1_ and A_2_ represent the amplitude of first exponential decay and second exponential decay, respectively. *t* is the independent variable, indicating the time at which the process occurs. From the dynamic results of these samples (Table 1), the decay time (τ_2_) of the ^4^S_3/2_ (540 nm) and ^4^F_9/2_ (654 nm) levels decreased significantly with the radii of alkali ions increasing, e.g., τ_2_ of ^2^H_11/2_ level decreased approximately by an order of magnitude, from 264.58 μs (Li^+^) into 27.07 μs (Cs^+^).

In the four samples, the kinetics was obviously changed; a comparatively short lifetime from CsLu_2_F_7_–Yb, Er nanoparticles suggests the distinct luminescence indicated that the increase in ions radii from Li^+^ (60 pm) to Cs^+^ (169 pm) ion altered the host lattice substitutionally or interstitially, which led to the difference of ETU population, owing to the symmetry of crystal field surrounding RE^3+^ ions changed. By comparison, the lifetime of Er^3+^ emission from Li/NaLuF_4_ nanoparticles was significantly longer, indicating the effective suppression of luminescence quenching. It could be attributed that the luminescence quenching caused by cross-relaxation would not predominate in these nanocrystals, owing to the recovery in the excitation energy.

Furthermore, the emission spectra were measured to examine the quenching effects induced by the concentration variation of Li^+^/Na^+^ co-doping nanoparticles. The data showed that an interesting UCL variation can be observed by increasing the doping ratio of Li^+^/Na^+^ concentration rather than considering the concentration quenching effect (Figure 5). As replaced by ions with larger ionic radii, the bond length between M–F (M = Li, Na) in the crystal structure would increase, which makes the crystal field intensity decrease and thus the emission peak splitting of Er^3+^ disappears. Notably, the population process of green emission is significantly suppressed as the ratio of Li^+^/Na^+^ is approximated [1].

This phenomenon could be attributed to the significant changes in the crystal structure caused by the co-doping of two kinds of alkali metals (Figure 6), thereby resulting in a series of crystal defects and then the suppression of the three-photon green emission. By comprehensive consideration, these results explicitly indicate that the defect-mediated energy transfer is responsible for the regulation of luminescence color and, thus, expands their optical applications. The ETU process of Tm^3+^ was also investigated (Appendix A, ESI). As shown in Figure 7, the luminescence performance of UCNPs in cells was detected by fluorescence microscopy imaging under near-infrared light irradiation. It has been demonstrated that UCNPs doped with four different ratios of alkali ions (Na^+^–Li^+^) exhibit significantly distinct imaging effects in cells. Amongst these, the signal intensity of the Na_0.9_Li_0.1_LuF_4_–Yb, Er nanoparticles was notably stronger than that of the other UCNPs. This phenomenon offers an effective means of obtaining fluorescent molecular probes with enhanced fluorescence emission. Therefore, the luminescence performance of rare-earth doped nanoprobes could be improved by optimizing the defect-mediated energy transfer by adjusting the lattice occupancy of alkali ions.

A similar ETU mechanistic investigation was corroborated by theoretical calculations, showing that different coordination forms could result in remarkable changes in the energy transfer between sensitizer ions and luminescence ions [39,40,41,42]. The site occupancy of different alkali ions could bring about different energy transfer paths, as shown in Figure 8. In the lattice structure of hexagonal UCNPs, there were cation sites selectively occupied by alkali ions and RE^3+^ ions, resulting in partial electron cloud distortion to accompany the lattice changes. When the ionic radii of alkali ions exhibit more differences from that of RE^3+^ ions, there will be more defects in the nanocrystals, which could shorten the lifetimes of the population and depopulation for corresponding energy levels; therefore, the UCL.

## 4. Conclusions

In this work, the doping concentration of Yb^3+^ and Er^3+^ fixed at 20% and 2%, the UCL properties were studied systematically for their influence on the energy transfer upconversion mechanism. The MLuF_4_-host (M: Li, Na) and MLu_2_F_7_-host (M: K, Cs) materials are synthesized through the solvothermal method. The steady-state PL showed a significant difference in visible-NIR range among the UCNPs with different alkali ions. The mechanism of energy transfer upconversion of these UCNPs was further explored to elucidate the influence of different alkali ions on crystalline structure through using the nanosecond time-resolved PL spectra. The change in ion radius from Li^+^ to Cs^+^ ion made the host lattice substitutionally or interstitially, which could alter the symmetry of the crystal field around RE-doped ions, leading to the improvement of UCL properties. The UCL properties should be attributed to the distortion of local asymmetry surrounding Er^3+^ ions. We also studied the luminescent properties of UCNPs containing different luminescent centers Er^3+^ and Tm^3+^. Our study highlights the influence of alkali ions on the ETU mechanism to improve the limitation of site occupancy in the upconversion nanomaterials for promoting their future applications.

## Data Availability

Data are contained within the article and Appendix A.

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
