# Peer review of "Defect-Mediated Energy Transfer Mechanism by Modulating Lattice Occupancy of Alkali Ions for the Optimization of Upconversion Luminescence"

_nanomaterials, 2024, doi:10.3390/nano14231969_

Round 1
Reviewer 1 Report
Comments and Suggestions for Authors
First of all, the manuscript does not appear to be adequately prepared for review. It requires significant attention to detail, especially in figures and their corresponding captions, which need careful verification and revision. The concept of defect-mediated energy transfer is not clearly explained. Comparing the UCL properties across samples with variations in crystal structure and particle sizes appears to be challenging. The fitting of the decay curves is overly simplified. The rising component should be added.
Author Response
We would like to sincerely thank you for your thoughtful and constructive comments on our manuscript. Your feedback has been invaluable in helping us improve the quality of our work, and we truly appreciate the time and effort you invested in reviewing our manuscript. We have carefully verified and modified figures and their corresponding captions. And based on the suggestions of the reviewers, the content of the article has been revised and improved. At the same time, we also added a rising component to the fitting of the attenuation curve, and the specific fitting results are shown in Table S1.
Reviewer 2 Report
Comments and Suggestions for Authors
The authors prepared the upconverting nanoparticles (UCNPs) doped different alkali ions (MLuF4 (M: Li, Na); MLu2F7 (M: K, Cs)) by conventional solvothermal method. In this case the doping concentration of Yb3+ and Er3+ was fixed at 20% and 2%. The authors also synthesized UCNPs doped with Yb3+ (20%) and Tm3+ (1%). The nanoparticles were characterized by TEM, EDX, XRD, steady-state upconversion luminescence spectra and time-resolved luminescence spectra. The influence of doping ions on the energy transfer upconversion mechanism in different defect crystals was systemically studied by changing the matrix alkali metal ions. Imaging of UCNPs in cells was demonstrated using a fluorescence microscopy under near-infrared light irradiation. I suggest that this manuscript be accepted for publication in Nanomaterials after addressing the following issues:
1. The authors imaged nanoparticles in cells in their work, but did not mention it in the abstract. This needs to be supplemented. Please also add in keywords: “imaging in cells”. Please supplement the information in the work on which cell line the research was conducted and the details of the experiments regarding the introduction of nanoparticles into cells (what concentrations of nanoparticles; in what solvent were the nanoparticles suspended; what number of cells, etc.)
2. Lines 95-96: the sentence “LiOH, KOH, CsOH·H2O were purchased from Alfa Aesar” is written twice. Please correct it.
3. Line 100: the authors omitted information about the LysoTracker dye, please complete.
4. Line 104: the authors did not describe the details of the synthesis regarding the doping of the nanoparticles with Yb and Tm. Please complete this information.
5. Line 111: in what volume of ethanol and cyclohexane were the particles purified, what were the centrifugation parameters (time, temperature, rpm). Please complete.
6. Line 114: nanoparticles were dried at 60oC, for how long? Please complete.
7. Experimental methods, Line 126: in the steady-state upconversion luminescence spectra experiments, were dried powders or solutions measured, and if solutions, in which solvent? What was the concentration of nanoparticles? What was the power density of the 980 nm laser used in these measurements? Was it the same for each sample? Please complete this information in your paper.
8. Line 134: when taking fluorescence microscopy images exposed by an 980 nm, what type of laser was used (pulse or continuous wave)? What was the laser power density? Were any filters used? Please complete this information in your paper.
9. Line 136: the name “LysoTracke” is incorrect, it should be LysoTracker. Please provide details on exactly what dye it was and describe the methodology for staining cells with both Hoechst and LysoTracker.
10. Line 144: “nanopar-ticles” spelling is incorrect, it should be “nanoparticles”.
11. Line 156: the authors must add under Scheme 1, that nanoparticles were doped with Er3+ and Yb3+ and with different alkali ions. Was the photo in Scheme 1 taken through a filter?
12. Line 160: the authors use a 980 nm laser for steady-state photoluminescence spectroscopy measurements. What was the laser power density? Were the measurements made in solution? What solvent? What nanoparticle concentrations?
13. Line 162: missing space: “[…] attributed to […]”.
14. Line 163: the authors write about luminescence at 1530 nm, but it is nowhere to be found in the spectrum. So there is no point in writing about it. Please take that into account.
15. Line 169: the entry is incorrect, it should be: “gradually”
16. Line 171: the authors need to enlarge the elements symbols in the EDX spectrum, because nothing can be seen in this form.
17. Line 179: the authors write about nanoparticles doped with Yb and Tm. Why were the nanoparticles doped with 1% Tm3+?
18. Line 179 and supplementary materials: in the captions under the graphs it should be written that these are nanoparticles with Tm and Yb and what was the power density of the 980 nm laser (Fig. S1, Fig. S2, Fig. S3, Fig. S4).
19. Line 188: there is a missing space: “[…] there with […]”; and "could" is written twice.
20. Line 202: please explain in the text what “A1”, “A2” and “t” are, because it was omitted.
21. Line 205, Figure 3.: it is not clear to the reviewer how the presented spectra differ from each other? Maybe in the concentrations of dopants? Please supplement this in the description of the graphs, you can use the notation: a, b, c, etc.
22. Line 216, Figure 4.: the authors signed the figure incorrectly. These are not steady-state measurements but kinetic curve and fitting lines. Please correct this in your work. For what concentrations of dopants are these graphs? This needs to be completed in the work because it is not clear.
23. Line 227: this paragraph starts with a different font, please correct it in the manuscript.
24. Line 236, Figure 5.: the y scale in graph 5 is not clear to the reviewer. Furthermore, the font on the y-axis is too small. Please also expand the notation MLuF4: Yb, Er, because in this form it is illegible.
25. Line 241: Figure 6. shows diffractograms, and the authors write about the steady-state PL luminescence. This needs to be corrected, because in this form it is unacceptable in the manuscript. The quality of the enlarged diffractograms on the yellow field also needs to be improved. Please fill in the caption whether these are nanoparticles doped with Er and Yb or Yb and Tm?
26. Lines 249-260: the authors conducted studies on cells, but it is not known what cell line it is, what concentrations of nanoparticles, incubation time, etc. What is the power of the 980 nm laser in the nanoparticle measurement channel and what in the case of Hoechst and LysoTracker? LysoTracker in the drawing, please correct the form of notation. What are the excitation and detection wavelengths? This must be completed in the manuscript.
27. Line 259: what do the stars in Figure 7b mean? Please fill in the caption under the graph with: a, b, etc.
28. Lines 265-269 are repeated twice in the manuscript, please correct it.
29. Figure 8.: Orange inscriptions on yellow background (ETU?) are not visible, please correct this. The caption under Figure 8. also needs to be changed because all nanoparticles are doped with Er and Yb and not just MLuF4: Yb, Er.
30. Conclusions: according to the reviewer, the results concerning the doping of Tm3+ and Yb3+ nanoparticles should also be commented on in the summary.
31. Supplementary materials, Figure S5.: the 1G4 (477 nm) what is this band? The reviewer does not see it on the Er3+ energy level diagram?
32. Supplementary materials, Figure S9.: what does BET stand for? And where is it visible on the Yb-Tm energy level diagram?
Round 2
Reviewer 2 Report
Comments and Suggestions for Authors
Line 138: The authors used a very high power density of a 980 nm laser, 500 W/cm2 for the measurements? What was the spot size and power in the measurements?
Line 217: Figure 3: Why did the authors doped the nanoparticles in Figure 3a with 20% Yb3+, 2% Er3+ and in Figure 3b with 20% Yb3+, 20% Er3+? Please explain in the text because it is not clear to the reviewer.
Line 245, the font needs to be improved
Line 259: The authors did not respond to the reviewer's question, what do the stars in Figure 7b mean? Please fill in the caption under the graph with: a, b, etc. What were the excitation and detection wavelengths used when taking the images?
Author Response
Comment 1: Line 138: The authors used a very high power density of a 980 nm laser, 500 W/cm2 for the measurements? What was the spot size and power in the measurements?
Response 1: Thank you for this comment. I'm sorry, I wrote the unit of power density incorrectly. The correct power density is 500 mW/cm2, and I have corrected it in the manuscript. The spot size and power in the measurement was 0.8 cm and 1 W, respectively
Comment 2: Line 217: Figure 3: Why did the authors doped the nanoparticles in Figure 3a with 20% Yb3+, 2% Er3+ and in Figure 3b with 20% Yb3+, 20% Er3+? Please explain in the text because it is not clear to the reviewer.
Response 2: Thank you for this comment. The two kinds of Er3+-doped nanoparticles were used to demonstrate that the defect-mediated energy transfer mechanism could be adjusted by the lattice occupancy of different alkali ions, especially for the impact of red light (4F9/2 → 4I15/2, 654 nm). And it has been explained in the manuscript.
Comment 3: Line 245, the font needs to be improved.
Response 3: Thank you for this comment. We have completed the correction in manuscript.
Comment 4: Line 259: The authors did not respond to the reviewer's question, what do the stars in Figure 7b mean? Please fill in the caption under the graph with: a, b, etc. What were the excitation and detection wavelengths used when taking the images?
Response 4: Thank you for this comment. The star represents that the UNCP has strong cell imaging performance.
One star represents that the UCNP cell imaging performance is relatively good, and three stars represent that the UCNP cell imaging performance is the best. If there are no stars, it means that the imaging performance of the UCNP cells is poor.
I have filled in the caption under the graph with “Fluorescence microscopy images of cells incubated with medium containing (a) NaLuF4: Yb, Er, (b) Na0.9Li0.1LuF4: Yb, Er, (c) Na0.5Li0.5LuF4: Yb, Er and (d) Na0.1Li0.9LuF4: Yb, Er nanoparticles under NIR laser radiation and the corresponding intracellular UCNPs fluorescence intensity.”
The excitation and detection wavelengths used for taking the images are 980 nm and 540 nm, respectively.
Round 3
Reviewer 2 Report
Comments and Suggestions for Authors
The author not correct the laser power density in supplementary materials under the Figure.
Author Response
Thank you for this comment. I have corrected the laser power density in Figures S1, S2, S3 and S4 in the supplementary materials. The modified content has been highlighted in yellow.